# The second Venus flyby of BepiColombo mission reveals stable atmosphere over decades

Jörn Helbert [1] ✉, Rainer Haus[1], Gabriele Arnold [1,2], Mario D'Amore[1], Alessandro Maturilli[1], Thomas Säuberlich[3] & Harald Hiesinger [4]

Studies of the Venusian mesosphere provide important information about the current state of the entire Venusian atmosphere. This includes information about the dense cloud structure, its vertical thermal profile, temperature fields, and the resulting dynamical and meteorological processes that contribute to a deeper understanding of the climatologically different evolutionary paths of Earth and Venus. However, the last measurements were acquired in 1983 during Venera-15 mission. In this paper, results of mid-infrared spectral measurements of the Venusian atmosphere are presented. Here we show Mercury Radiometer and Thermal Infrared Spectrometer (MERTIS) measurements of the Venusian atmosphere during the second flyby of BepiColombo mission on its way to Mercury. Our Venus measurements provide reliable retrievals of mesospheric temperature profiles and cloud parameters between 60 and 75 km altitude, although MERTIS was only designed to operate in Mercury environment. Our results are in good agreement with the Venera-15 mission findings. This indicates the stability of the Venusian atmosphere on time scales of decades.

With Venus and Earth we have two planets in the solar system that formed in the same region of the solar nebula have similar size, but evolved dramatically different. While Earth developed into a habitable planet, Venus took a divergent development path. Understanding the divergent evolution of Earth and Venus especially in the context of potentially Earth like exoplanets is major challenges in planetary and solar system research. Any knowledge increase in this context will bring forward our understanding of possible feedback mechanisms in the sensitive Earth's climate system[1].

Analyses of the thermal structure and cloud formation patterns are of central importance for assessing dynamical and meteorological processes and the environmental conditions prevailing on Venus. Especially the transition region between troposphere and mesosphere at about 60-70 km altitude is important for climate tracking, since it displays a strong variability with latitude and local time.

An overview of the most important past experiments on remote sensing of temperature altitude profiles and cloud features in the atmosphere of Venus has been given by Haus et al.[2]. Much more information originated from ESA's Venus Express mission (VEX, 2006-2014, Svedhem et al.[3]). In particular, the Visible and Infrared Thermal Imaging Spectrometer (VIRTIS, Drossart et al.[4]) and the Venus Radio science experiment (VeRa, Tellmann et al.[5]) on VEX provided large data amounts on global scales. Latest data came from the Japanese Akatsuki mission (2016-2018, Nakamura et al.[6]).

After its successful launch in October 2018 from the European Spaceport in Kourou, French Guiana, the ESA-JAXA mission BepiColombo is currently on its nominal 7-year journey to Mercury (Benkhoff et al.)[7]. The interplanetary cruise includes nine flybys for gravitational assists: one at Earth, two at Venus and six at Mercury. The Earth flyby and the first Venus flyby (FB1) took place in April and

[1]German Aerospace Center (DLR), Institute of Planetary Research, Rutherfordstrasse 2, 12489 Berlin, Germany. [2]University Potsdam, Institute of Geoscience, Karl-Liebknecht-Str. 27, 14476 Potsdam, Germany. [3]German Aerospace Center (DLR), Institute of Optical Sensorsystems, Rutherfordstrasse 2, 12489 Berlin, Germany. [4]University Muenster, Institute of Planetology, Wilhelm-Klemm-Str. 10, 48149 Muenster, Germany. ✉e-mail: joern.helbert@dlr.de

October 2020, respectively. The second Venus flyby (FB2), on which the present work focuses, took place on August 10, 2021 with a closest approach (CA) at 13:51 UTC at a minimum altitude of 552 km (Mangano et al.)[8].

In this paper, thermal infrared spectra obtained by the infrared (IR) grating spectrometer (TIS) of the MERTIS (MErcury Radiometer and Thermal Infrared Spectrometer) instrument during the second Venus flyby of the BepiColombo mission are presented and analyzed. The MERTIS observations provide not only temperature profiles but also independent information on $SO_2$ and $H_2SO_4$ cloud aerosol properties. The retrieved temperature profile at latitude 10°N well agrees both with the corresponding VIRA (Venus International Reference Atmosphere) profile and the profile obtained from the directly comparable Venera-15 PMV (Profile Measuring Instrument for Venus) data 37 years ago. Retrieved cloud properties (mode factors $MF_{1/2}$) and cloud top altitudes $z_t$ also well coincide.

## Results

Our Venus observations were performed with MERTIS, which is an instrument designed to operate in Mercury environment. As described in (Hiesinger et al.)[9] MERTIS combines a push-broom IR grating spectrometer (TIS) with a radiometer (TIR). TIS operates between 7 and 14 μm and will record dayside radiance spectra of Mercury to infer surface emissivity characteristics, whereas TIR is going to measure the surface temperature (80–700 K) at night- and dayside in the spectral range from 7 40 μm. TIR is implemented by an in-plane separation arrangement. In this configuration the two radiometer detector lines form the slit of the TIS channel which is an imaging spectrometer (Hiesinger et al.)[9]. It uses the first European-built space-qualified uncooled microbolometer array. The optical design of MERTIS combines a three mirror anastigmat (TMA) with a modified Offner grating spectrometer. A pointing device allows viewing the planet (planet-baffle), deep space (space-baffle), and two black bodies at 300 K and 700 K temperature, respectively.

The observations of Venus with MERTIS have been challenging. The planet Mercury exhibits dayside surface temperatures up to 700 K. Maximum infrared brightness temperatures that can be observed over the dense atmosphere of Venus are in the order of 220–260 K. This pushes the instrument to its sensitivity limits. Moreover, the team had to devise a new observation and calibration strategy to make these observations possible at all. During cruise the ESA Mercury Planetary Orbiter (MPO) and the JAXA Mercury Magnetospheric orbiter (MMO) fly in a composite configuration with a propulsion element, the Mercury Transfer Module (MTM) and a sunshade cone (MOSIF) to protect the MMO. In this configuration the nadir panel (z-axis) of the MPO points towards the MTM. Therefore, most instruments cannot operate during cruise. MERTIS has a viewport through the space baffle which is used in nominal operations for deep space calibration (Hiesinger et al.)[9]. During the Venus flyby this port was used for planetary observations. It means however that the instrument had to operate in a way that it uses the deep space calibration port as its main observation port while maintaining the standard sequence that includes regular observations of the internal calibration targets. Calibration using deep space, which is the key to obtain the thermal radiation coming from the instrument itself, was obtained before and after the flyby. The whole observing timeline and the instrument observation procedure had been tested during several instrument checkouts.

MERTIS theoretically has an accuracy in the range of 0.03 K based on its internal calibration when using the MERTIS planetary baffle, which is optimized to reject all out-of-field radiation from a source larger than the field of view of the instrument. For the Venus observations, however, the instrument had to observe using the space port. This port has a simpler, asymmetrically shaped baffle, which means that stray light from the extended disk of Venus must be

considered as an additional source of uncertainty. Since no in-flight calibration of the space port could be performed for the Venus flyby, we chose the data clustering described in the paper to minimize this error for temperature retrieval. The closer BC flyby of Venus, FB2, was explicitly suited for developing this methodology because nadir-equivalent transmitted atmospheric columns could be directly compared. A more detailed analysis of FB1 in a follow-up publication may be able to place additional constraints on the effect of stray light for a larger interval of nadir observation geometries. In this paper, we focus on the comparison with the atmospheric models and previous mid-IR nadir studies of the Venusian mesosphere as the PMV measurements. We discuss these results in comparison with recent high-precision temperature profiles obtained from radio occultation measurements[5,10–12].

Measurements with the MERTIS hyperspectral channel (MERTIS-TIS) are the first spectrally resolved observations of Venus in the thermal spectral range longward of 5 μm since the Venera-15 Fourier spectrometer experiment FS-1/4 in 1983 (Oertel and Moroz, 1984; Oertel et al.)[13,14]. This instrument (also named PMV, Profile Measuring Instrument for Venus) obtained spectra in the 6.0-36.5 μm range at spectral resolution of 6.3 cm⁻¹. The MERTIS resolution of 90 nm corresponds to 18 cm⁻¹ near 7 μm and 5 cm⁻¹ near 14 μm. This offers the great opportunity to further test and adapt software and to compare retrieval results for atmospheric parameters of Venus.

This paper elaborates on the similarities of MERTIS and PMV data. It is shown that the radiation measurements performed by the two experiments are well eligible to reliably retrieve mesospheric parameters of Venus like temperature profiles and cloud properties, and that the results largely agree. This contributes to our understanding of the atmosphere of Venus and its stability on timescales of decades. It is also one main goal of the present paper to demonstrate the capabilities of an instrument like MERTIS using an uncooled microbolometer for hyperspectral observations of relative cool objects as in case of Venus' mesosphere.

Present results create the prerequisites to investigate MERTIS-flyby 1 data that will be done in near future. The observation geometry of FB1 is more challenging due to the spaceship's larger distance to Venus. As a consequence, FB1 observations permitted much larger latitude coverage from 50°S to 85°N.

## Observations

MERTIS data analyses that are presented here are based on Venus flyby 2 measurements. Planetary radiation has been recorded through the MERTIS space baffle that was designed to perform deep-space calibration measurements at Mercury. Due to the spacecraft trajectory and the comparatively low distance to Venus (6300 km on average) most data were acquired at latitudes between 8 and 14°N and at local times between 7 and 15 h. Therefore, FB2 data contain limited information on Venus thermal structure on global scale from the outset, but nevertheless their analysis is important to validate the above discussed approach (direct comparison with PMV, preparation of FB1 data analyses).

During FB2 the spacecraft approached the planet from its night side, and the closest approach (CA) occurred near the evening terminator of the planet, but slightly shifted toward late afternoon side. Then the spacecraft moved away from the planet on the dayside. MERTIS observations were performed from −4 min to +23 min around CA, from distances ranging from 13,000 km to 900 km. This covered local times between 6.0 and 18.0 h at low latitudes. MERTIS obtained around 2,200 spectrometer channel measurements as well as about 300 measurements with the radiometer channel. The TIS channel was working at full spatial resolution without any spatial binning providing 100 pixels across the track, and >900,000 single spectra of Venus could be recorded in spite of the non-standard operation procedure.

## Data selection

Radiances of some individual spectra are very low, and absolute values between 6.8 and 8 μm and near 13.5 μm (near the edges of the measurement range) are close to zero in these cases. This is most likely due to a sometimes incomplete illumination of individual detector pixels (partly deep space signals). The calculation of brightness temperatures from very low radiances applying Planck's law leads to non-physical results. Therefore, those spectra are removed from the basic data archive. Remaining data involve about 94,000 spectra. Figure 1a illustrates the distribution of usable spectra with local time and latitude. Figure 1c shows a histogram of spectrum numbers as function of 10° observation angle (often denoted as emission angle) intervals.

It is important to estimate the range of possible brightness temperature ($T_B$) changes due to observation angle ($\upsilon_{obs}$) variation and to determine limits for angle variations that keep $T_B$ changes as small as possible[15], when spectra averaging is applied prior to retrieval calculations. $\upsilon_{obs}$ selection can be accomplished by determination of his-

tograms of spectrum distribution for 10° $\upsilon_{obs}$ intervals. Large $\upsilon_{obs}$ values imply long atmospheric paths and may result in much lower radiances and brightness temperatures compared to observations near nadir geometry. The interval for maximum spectrum population is then selected for further processing. Considering all latitudes in the belt (10 ± 2.5°), maximum $\upsilon_{obs}$ population occurs in the interval 20°–30° (see Fig. 1d) corresponding to local times between 10.5 and 13.0 h as shown in Fig. 1b for the corresponding remaining spectra (about 13,000). Measurements for angles >75° should not be used, since the assumption of plane parallel atmospheric layers is no longer valid, which is a requirement of the used radiative transfer model.

Figure 2a shows a selection of FB2 individual radiance measurements and the zonal average spectrum in the 10°N latitude belt applying the data selection steps described above. Since the instrument under Venus conditions has worked at its limits, single spectra are very noisy and inappropriate for use in retrieval procedures. Therefore, detailed error analyses of retrieval results are not possible.

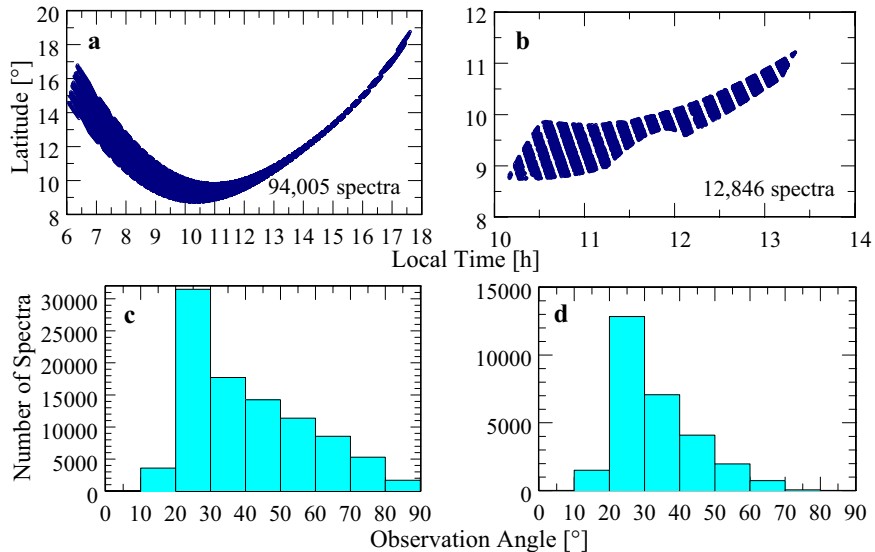

**Fig. 1 | MERTIS observations of Venus. a** Show all usable spectra, while **b** shows only spectra in the latitude belt (10 ± 2.5)° **c** shows the distribution of all spectra, while **d** shows the distribution of all spectra in the latitude belt (10 ± 2.5)°.

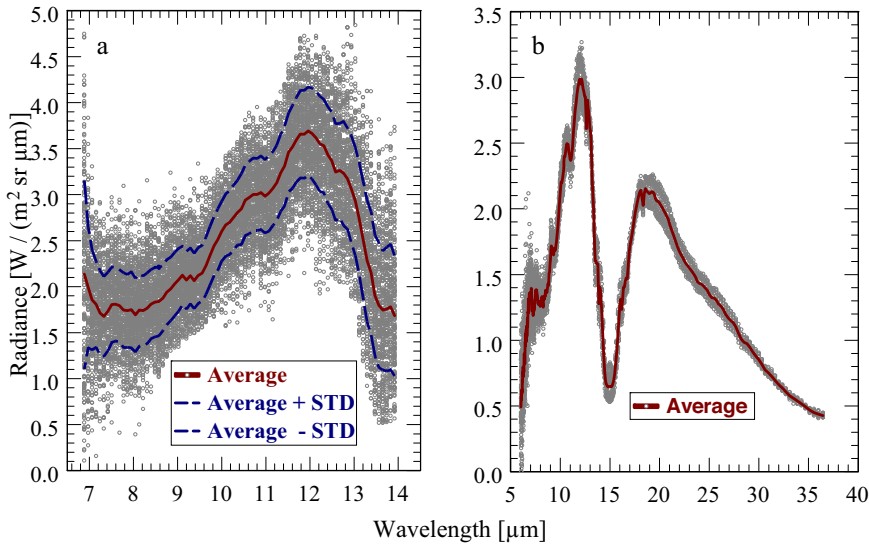

**Fig. 2 | Individual and averaged radiance spectra. a** MERTIS-FB2 near 10°N. **b** PMV near 30°N. Shown in both panels are the average spectra as well as the spectra plus/minus 1 standard deviation.

Averaging leads to a major reduction of measurement errors. This is proven by the structure of the averaged spectrum, which is well comparable with PMV.

It is an important goal of the present paper to compare atmospheric parameter retrieval results based on MERTIS-TIS and Venera-15 (PMV) measurements. PMV observed the northern hemisphere of Venus. Data cover the spectral range 275-1656 cm$^{-1}$ (6.0-36.5 μm) that completely includes the strong 15 μm $CO_2$ band, which is well suitable for mesospheric temperature retrievals. Most measurements were performed at latitudes between 30 and 85°N both over the night- and dayside of the planet at all local times (except 10.5-16.0 h). MERTIS data were exclusively collected on the dayside. Note that atmospheric emissions in the thermal spectral range longward of 7 μm are not affected by solar radiation (Haus et al.)[2]. PMV spectra population southward of 30°N is very sparse, and $v_{obs}$ values are mostly >60°. Figure 2b shows PMV individual radiance measurements recorded over the latitude belt (30 ± 2.5) °N and the resulting zonal average spectrum applying the same data selection steps as in case of MERTIS. Maximum $v_{obs}$ population occurs in the interval 50°–60° here. Atmospheric temperature altitude profiles at low latitudes (e.g. 10° and 30°) are known to be very similar. PMV radiances are smaller compared to MERTIS radiances due to much larger observation angles.

## Atmospheric temperature profiles

Weighting functions W that are derivatives of wavenumber-dependent atmospheric column transmittances above an altitude reference level are used for the atmospheric temperature profile retrieval algorithm[2]. They determine the effective altitude range of temperature sounding. Figure 3a illustrates weighting functions for PMV and MERTIS in the spectral ranges 300–1100 cm$^{-1}$ and 720–1100 cm$^{-1}$, respectively. The spectral resolution of PMV data is 6.3 cm$^{-1}$. The MERTIS resolution of 90 nm corresponds to 18 cm$^{-1}$ near 7 μm and 5 cm$^{-1}$ near 14 μm. Specific profiles of W depend on latitude (different temperature profiles), cloud optical thickness and observation angle. The used cloud model is described in the Methods section. W's in Fig. 3a are based on VIRA temperatures at 45°, standard cloud model and nadir observation geometry. Secondary W maxima below 60 km appear at some wavenumbers that may exceed the magnitude of maxima at higher altitudes. The secondary maxima are exclusively due to cloud influence. The retrieval algorithm uses the full W profiles that yield some temperature information up to ±5 km beyond the $W_{max}$ altitude, however maximum temperature information at a certain wavenumber

originates from the altitude where the weighting function maximum ($W_{max}$) is located. Figure 3b shows the altitudes of $W_{max}$ as function of wavenumber in the PMV (open circles) and MERTIS (solid boxes) measurement ranges based on the plots in Fig. 3a. Note that only each third data point is marked in Fig. 3b. The depressions of $W_{max}$ near 830 cm$^{-1}$ are due to the secondary maxima in the weighting functions.

Figure 3b highlights a limitation of MERTIS data compared to PMV measurements. In case of PMV, the 15 μm $CO_2$ band (centered at 667 cm$^{-1}$) and its wings sound the altitude range of about 55–90 km. In case of MERTIS, the lower wavenumber boundary is located at about 720 cm$^{-1}$. Large parts of the 15 μm band are not covered by MERTIS, therefore. This instrument was constructed to collect surface spectra of Mercury. It was not optimized for atmospheric measurements on Venus. MERTIS data retrieval procedures are mainly sensitive to an altitude range of 60-75 km.

It is an important prerequisite of MERTIS data analyses to demonstrate that useful information on Venus' atmospheric temperature profiles and cloud parameters can be still extracted from available spectral information. For this purpose, retrievals from PMV measurements were performed at the beginning that utilize two different spectral ranges. Range A extends from 300 to 1100 cm$^{-1}$ (9.1–33.3 μm). The lower wavenumber boundary of range B is determined by the first usable MERTIS data point at about 720 cm$^{-1}$, that is, retrievals are restricted to the range 720-1100 cm$^{-1}$ (9.1–13.9 μm). Figure 4a compares zonal averages of measured (empty circles) and simulated PMV brightness temperature spectra at 30°N using ranges A and B in the retrieval procedure, respectively. Range B results are marked by empty boxes. Simulated spectra are obtained from forward modeling using the retrieved parameters as input quantities. The agreement between measurement and simulation is very good for both ranges. This implies that atmospheric parameters retrieved from A and B should largely agree. Figure 4b displays the retrieved PMV temperature profiles TA(z) and TB(z). The initial profile TI(z) (VIRA 30°, empty circles) is also shown for comparison. The inset of Fig. 4b displays the temperature difference TD(z) where TD(z) = TB(z)-TA(z). In general, range A and B profiles are very similar. Maximum differences between 61 and 76 km do not exceed 1 K. Larger deviations (typically higher TB values) occur above 76 km and below 61 km. Deviations at high altitudes are simply due to the fact that range B retrievals are not sensitive to this altitude domain. The differences between 55 and 61 km result from the significantly smaller number

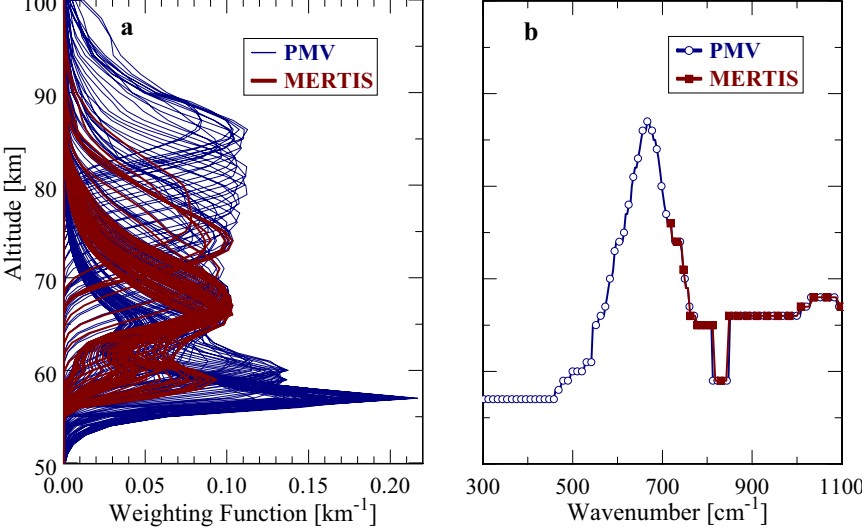

**Fig. 3 | Due to the different spectral coverage and spectral resolution MERTIS and PMV have different weighting functions. a** Shows the spectral weighting functions for MERTIS and PMV. **b** Shows the altitudes of the weighting function maxima as function of spectral channel for MERTIS and PMV.

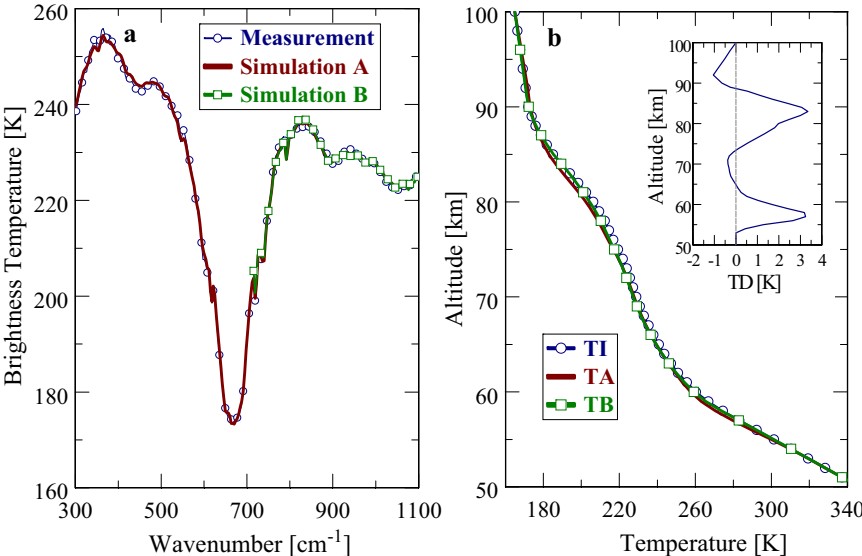

**Fig. 4 | Comparison of the PMV brightness temperature spectra with simulation and of the initial and retrieved temperature profiles. a** Comparison of zonal averages of measured and simulated PMV brightness temperature spectra at 30°N. **b** Comparison of initial and retrieved temperature profiles at 30°N. TI: Initial profile (VIRA 10–30°N), TA: Retrieved profile for spectral range A (9.1–33.3 μm), TB: Retrieved profile for spectral range B (9.1–13.9 μm). Inset: TD Temperature difference TB-TA.

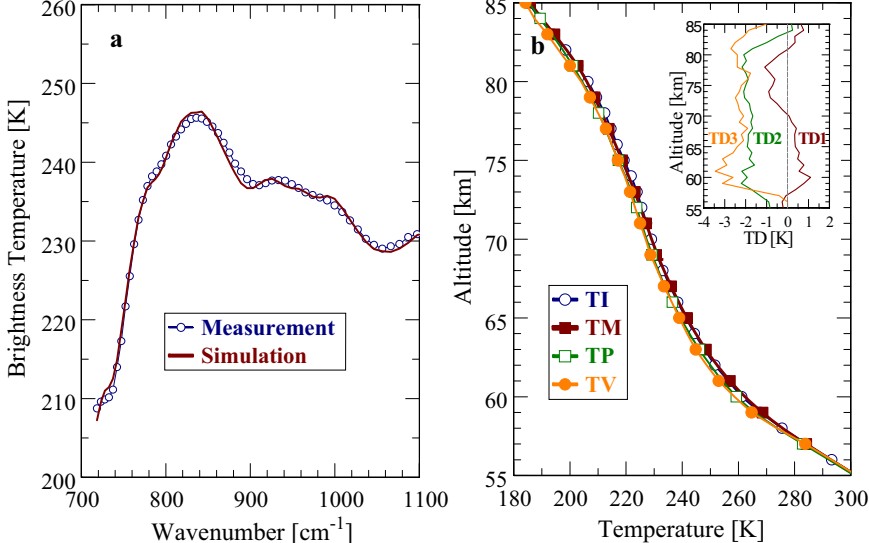

**Fig. 5 | Same as Fig. 4 for the MERTIS instrument—comparison of MERTIS brightness temperature spectra with simulation and of the initial and the retrieved temperature profile, as well as a comparison with retrievals from previous experiments. a** Comparison of zonal averages of measured and simulated MERTIS brightness temperature spectra at 10°N. **b** Comparison of initial and retrieved temperature profiles. TI: Initial profile (VIRA 10-30°N), TM: Retrieved MERTIS profile at 10°N. TP: Retrieved PMV profile at 30°N. TV: Retrieved VIRTIS profile at 10°N. Inset: TD1: Temperature difference TM-TI. TD2: Temperature difference TP-TI. TD3: Temperature difference TV-TI.

of weighting channels in range B compared with A. Retrieved cloud mode factors of modes 1 and 2 ($MF_{1/2}$, see Methods section) and cloud top altitudes $z_t$ from both ranges are identical, $MF_{1/2} = 0.93$, $z_t(10\,\mu m) = 66.4$ km. Factors of modes 3 and 4 cannot be retrieved in the used spectral ranges and are set to unity.

These comparisons of retrieval results from PMV data obtained from different spectral ranges are encouraging with respect to MERTIS data. Use of spectral information from the long wavenumber wing of the 15 μm band (720–1100 $cm^{-1}$, 9.1–13.9 μm) should provide reliable information on temperature profiles in the lower mesosphere (60–75 km) of Venus.

Figure 5a compares zonal averages of measured and simulated brightness temperature spectra at 10°N obtained from MERTIS FB2 data. Zonal averages extend over local times between 10.5 and 13.0 h. The agreement of curves is quite good and well comparable to PMV results. Deviations of brightness temperatures ($|\Delta T_B|$) are mostly <1 K and never exceed 1.5 K. The retrieved temperature profile TM(z) is illustrated in Fig. 5b. The line marked by empty circles describes the VIRA profile TI(z) for low latitudes (0-30°), which is used as initial model in the retrieval algorithm. Atmospheric temperature profiles at equatorial and low latitudes are known to be very similar. The PMV retrieval result (range B) for 30°N (empty boxes), TP(z) is shown for

comparison. As mentioned above, reliable PMV data are not available southward of 30°N. The line marked by solid boxes displays a retrieval result TV(z) using VIRTIS data (Haus et al.)[15]. The agreement of retrieval results obtained from the three different experiments is very good. All temperature profiles well coincide with the VIRA model. The inset of Fig. 5b displays the temperature differences TD1(z), TD2(z) and TD3(z) where TD1(z) = TM(z)-TI(z), TD2(z) = TP(z)-TI(z), TD3(z)=TV(z)-TI(z). Maximum differences between the MERTIS and VIRA temperature profile (TD1) are in the order of 1 K. Note that the determination of simulated $T_B$ spectra that optimally fit the measurements is the only decisive condition in the retrieval procedure, albeit retrieved temperature profiles are forced to approach the used initial model at altitude regions where temperature weighting functions become very small or even zero due to missing spectral information (e.g. at altitudes higher than about 77 km). Retrieved cloud parameters from MERTIS FB2 data at 10°N are $MF_{1/2}$ = 0.96 and $z_t$ (10 μm) = 66.5 km that represent typical values near the equator and that well agree with PMV and VIRTIS results.

The present paper demonstrates that mesospheric parameters of Venus like temperature altitude profiles and cloud properties can be reliably determined from BepiColombo MERTIS spectra recorded during flyby 2. MERTIS sensed Venus' middle atmosphere and cloud layers at altitudes between about 60 and 75 km. The MERTIS observations are the first hyperspectral mid infrared data in four decades obtained close to Venus. They complement the radio occultation observations performed by Venus Express and Akatsuki (Tellmann et al.[16], Piccialli et al.[10], Imamura et al.[11], Ando et al.[12]). The MERTIS observations independently confirm the remarkable overall stability of the atmosphere especially in the equatorial region. They are complementary to the radio occultation observation by PMV, PVO, Venus Express, Magellan and Akatsuki observations (see Fig. 6 in Imamura et al.[11] as well as Fig. 4a in Ando et al.[12]) which are sensitive to smaller scale variations.

The MERTIS observations provide not only temperature profiles but also independent information on $SO_2$ and $H_2SO_4$ cloud aerosol properties. The retrieved temperature profile at latitude 10°N well agrees both with the corresponding VIRA profile and the profile obtained from the directly comparable Venera-15 PMV data 40 years ago. Retrieved cloud properties (mode factors $MF_{1/2}$) and cloud top altitudes $z_t$ also well coincide.

Due to the spacecraft trajectory and the comparatively low distance to Venus, the majority of FB2 data was acquired at over the narrow latitude belt around 10°N. Therefore, information on atmospheric parameters on global scale is not available from the outset. Methods and results of MERTIS-FB2 data analyses provide a valuable base for current investigations of FB1 data. Due to significantly larger distances to Venus FB1 covered a much broader range of latitudes (50°S to 85°N). This will allow to study two-dimensional mesospheric temperature fields and to compare the results with findings from previous experiments (PMV[13,14], VIRTIS[4]) as well as a comparison with the data provided by the TIR imager on the JAXA Akatsuki mission.

The confirmation of temperature profiles in the atmosphere of Venus is important for the upcoming orbital mission to Venus, especially the ESA EnVision and NASA VERITAS missions. Both missions carry the Venus Emissivity Mapper (VEM), a near infrared instrument that will provide for the first time a global map of surface rock types from orbit. In order to be able to retrieve this information the data analysis algorithm of VEM uses a radiative transfer model to relies on the accuracy of our understanding of the atmosphere of Venus. In addition, Envision carries a suite of three spectrometer covering the UV to NIR range with the goal to provide a holistic view of the coupled system of surface and atmosphere on Venus. This suite and especially the coupling between the channels in the suite again rely on a baseline profile of the atmosphere and any additional data point that allows to confirm the currently used models will benefit the development of this mission.

Given the lack of orbital spectral data from Venus, each new data set from space-borne experiments near this planet in the neighborhood of our home planet Earth is very important to facilitate studies of comparative planetology and a better understanding of possible feedback mechanisms in the sensitive Earth's climate system. MERTIS observations and data analyses contribute to our understanding of the atmosphere of Venus and its stability on timescales of decades. They can also contribute to study atmospheric structure, cloud level chemical processes, radiative energy balance, and to understand the impact of global-scale atmospheric waves on Venus' weather patterns.

At the same time, present results highlight the capabilities of the MERTIS instrument specifically and more general the usability of uncooled microbolometers for hyperspectral observations of relative cool objects. MERTIS-TIS was solely designed to observe the hot surface of Mercury at temperatures up to 700 K. Much lower temperatures in the mesosphere of Venus (220-260 K) push the instrument to its sensitivity limits. It is shown here, however, that reliable parameters of Venus' atmosphere can be determined from such kind of measurements. This opens very promising future prospects for a new generation of imaging MIR spectrometers without a lifetime-limited actively cooled detector array, especially in the context of future missions for long-term studies of the middle atmosphere of Venus. None of the currently selected missions carries such an instrument. With the results presented here we encourage the inclusion of such an instrument based on a microbolometer detector in future mission concepts.

## Methods
### Data calibration
The complete data acquisition and processing architecture is described in detail by D'Amore et al.[17]. As a high-level view, MERTIS telemetry data are transmitted from the spacecraft to the BepiColombo Mission Operations Center, then transmitted to the ESA Science Ground Segment. The MERTIS Team at DLR has developed and maintains the data processing pipeline to move from telemetry to raw and finally to calibration and derived data. The data are then distributed back to the Team and to ESA's Planetary Science Archive (PSA) for the public distribution. MERTIS produces sensor data (TIS & TIR) and metadata describing the instrument status and environment, like detector temperatures etc.

The housekeeping data are sent already calibrated in physical units from onboard firmware, without needing for further processing. Only in case of in-flight discovery of discrepancies an extra step will be implemented to correct for spurious effects. The radiometer and spectrometer calibration follow the same logic: A nominal observation sequence starts with setting sights on the internal cold and hot blackbodies, followed by the space view pointing and then a succession of planetary observations at varying time interval, and again the blackbodies and space view cycle.

MERTIS-TIS was characterized in laboratory in a radiometric calibration campaign and the radiometric system response measurements under space-like thermal-vacuum conditions[9,17]. The measurements include the spectral assignment for each spatial/spectral channel and determination of the dynamic range limits of the image signal in order to keep a mostly linear system response. Furthermore, they include the definition of different sets of detector settings in order to ensure the system's ability to deliver high quality images for the whole temperature interval that the instrument will face in orbit operation around Mercury. An advanced calibration approach has been introduced using floating sensitivity coefficients in order to compensate for residual non-linearities of the image signal within the configured dynamic range. Together with correlated image noise removal algorithms (image offset noise, stripe noise) the radiometric calibration delivers very reproducible and accurate results even under strong temperature changes of the instrument during the mission.

Venus observations present a special challenge for the instrument, because observed brightness temperatures are much lower than Mercury dayside temperatures. The calibration procedure was enhanced to take into account deep space pre- and post- Venus spectra and adjust the image offset and stripe noise accordingly with this special observation. MERTIS has two internal blackbodies, one at instrument temperature (equipped with two redundant PT1000 temperature sensors) and the second actively heated to 700 K. During the observations the instrument every 60 s takes observations of both blackbodies to verify the calibration. During the Venus flyby the first blackbody exhibits temperatures between 282.4 K and 286.9 K as measured by the temperature sensors embedded in the blackbody surface. This is slightly higher but still comparable with range of temperatures observed at Venus. The regular calibration measurements show a difference of <0.03 K between measured blackbody temperatures and temperatures retrieved from the spectral measurements.

The geometric registration uses NAIF/SPICE kernels to characterize MERTIS-TIR and -TIS field of view on the target surface. To date, the kernels describing MERTIS are in the process of being updated and validated using the Near-Earth Commissioning and Moon flyby data. This work uses already kernels reflecting those more precise geometric characterization available to date. This includes validation of the internal alignment using the onboard 700 K blackbody emitter as well as validation of the external alignment based on imaging of the Moon during the Moon flyby. The calibrated and geometrically registered data blocks are encoded using the PDS4 template for the CAL data files.

## Observation planning and execution

Since the line of sight in the nadir direction for most instruments on the BepiColombo MPO is blocked by the Mercury Transfer Module (MTM) during cruise and flyby operations (including the MERTIS planetary baffle), the standard acquisition procedure for MERTIS was modified to allow acquiring spectra of flyby targets through its space baffle. In fact, the MERTIS pointing device allows viewing the planet (planet-baffle), deep space (space-baffle), and two internal black bodies at 300 K and 700 K temperature, respectively. The MERTIS operations software was adapted to allow for this unique opportunity. With the Earth/Moon flyby the MERTIS imaging spectrometer provided the unique hyper-spectral observation of the Moon in the thermal infrared wavelength range from space. During the successive two Venus flybys MERTIS performed observations of the Venus atmosphere down to about 60 km altitude in the thermal infrared, 37 years after the last available remote sensing data from space at these wavelengths.

The fact that the observations passed from the nightside to the dayside added another challenge to the observations. The radiator of the spacecraft had to be pointed to Venus to allow the observations with the MERTIS calibration port. Transitioning from nightside to dayside resulted in an increase in the instruments baseplate temperature by 3.2 K and detector housing temperature by 4.2 K. However, the thermal control system of the microbolometer kept the chip temperature stable within a range of 23 mK (Helbert et al.[18]; Walter et al.[19]; Peter et al.[20]).

## Radiative transfer model

A radiative transfer simulation and retrieval algorithm that was originally developed for PMV data analyses by Haus et al.[2] has been adapted to investigate MERTIS-TIS data and to extract atmospheric temperature altitude profiles and cloud parameters of Venus' lower mesosphere (60-75 km). The radiative transfer model (RTM) includes the discrete ordinate package DISORT (Stamnes et al.)[21]. It considers absorption, emission, and multiple scattering by gaseous and particulate atmospheric constituents. Look-up tables of quasi-monochromatic absorption cross-sections of atmospheric gaseous constituents are calculated on the basis of a line-by-line procedure for

a variety of temperature and pressure values being representative for Venus' atmosphere at altitudes from the surface up to 140 km, and for spectroscopic parameters, which were specified by Haus et al.[2]. $CO_2$ is the main gaseous constituent. It has a constant volume mixing ratio of 96.5%. Minor constituent contributions ($H_2O$, $SO_2$) may be neglected in the spectral range used for MERTIS retrieval calculations.

The initial cloud model facilitates analytical descriptions of four-modal particle altitude distributions (Haus et al.)[2] where the modes M1, M2, M2' and M3 are assumed to consist of spherical $H_2SO_4$ aerosols at 75 weight-% solution. Mie scattering theory (Wiscombe.)[22] is applied to derive wavelength-dependent microphysical parameters of each mode (absorption, scattering and extinction cross-sections, single scattering albedo, asymmetry parameter, and phase function). Log-normal size distributions are used with modal radii of 0.3, 1.0, 1.4, 3.65 μm and dimensionless dispersions of 1.56, 1.29, 1.23, 1.28, respectively (Pollack et al.)[23]. Refractive index data are taken from Palmer and Williams[24]. The strongly changing optical depth of Venus' clouds is modeled by variation of individual mode particle number densities. Introducing so-called altitude-independent cloud mode factors $MF_j$, the altitude-dependent layer optical depth of mode j may be expressed by $u_j(z) = MF_j D_j(z) \beta_j$ where $D_j$ is the particle number density integral over an atmospheric layer centered at z, and $\beta_j$ (in units of cm2) the extinction cross-sections of mode j, respectively. Thus, the number density of particles can be varied independently for each cloud mode, but maintaining the altitude distribution that is determined by the initial model. For all $MF_j = 1.0$ and the reference wavelength 10 μm, the total cumulative cloud optical depth (cloud opacity) and the cloud top altitude are 28.34 and 66.64 km. Cloud top altitude is defined as the altitude where the cumulative optical depth (COD) becomes unity.

## Retrieval technique

The retrieval used in this work is discussed in details in Haus et al.[15]. In order to extract atmospheric temperature profiles and cloud mode factors $MF_{1/2}$ from the measurements, the RTM is embedded into a retrieval technique that makes simultaneous use of information from different spectral ranges of an individual spectrum. The retrieval algorithm iteratively optimizes the parameters until the simulated radiance spectrum well fits the measurement for all utilized spectral ranges in the least-squares sense. The determined parameters are interpreted to represent the state of atmosphere that led to the observed spectrum. Unfortunately, the retrieval technique is not able to disentangle mode M2' and mode M3 influences from mode M2 and M1 changes, and $MF_{2'}$ and $MF_3$ are always set to unity, therefore.

Mesospheric temperature profiles are retrieved using Smith's relaxation method (Smith.)[24]. It employs initial and measured brightness temperatures in the 15 μm $CO_2$ band. Initial latitude-dependent temperature structure models are taken from the Venus International Reference Atmosphere (VIRA) (Seiff et al.[25]; Zasova et al.[26,27]). The temperature is retrieved as a function of altitude z. Since atmospheric pressure p is intrinsically the basic quantity for temperature calculation, p(z) is recalculated in each retrieval step using temperature and atmospheric scale height from the previous one. Initial value in the barometric formula is always the surface pressure (92.1 bar on Venus).

The method of self-consistent temperature profile and cloud parameter retrievals from MERTIS and PMV spectra encompasses five steps:

1. Preliminary temperature profile T1(z) at 720–1100 cm$^{-1}$ for initial cloud model ($MF_j = 1$, $z_t$ (10 μm) = 66.6 km);
2. Cloud mode factor $MF_{1/2}$ and cloud top altitude $z_t$ for T1(z) at 830–1100 cm$^{-1}$;
3. Temperature profile T2(z) at 720–1100 cm$^{-1}$ using $MF_{1/2}$ and $z_t$ from step2;
4. Iteration of steps 2 and 3;
5. Simulation of final brightness temperature spectrum $T_B(v)$ for all new parameters over the full MERTIS / PMV spectral range.

## Data availability

The datasets generated during and/or analysed during the current study are available from the corresponding author upon request. The spectrometer data used in this study are available from the ESA Planetary Science Archive (https://archives.esac.esa.int/psa/#!Table%20View/MERTIS=instrument) following the ESA BepiColombo data release cycle for calibrated data. All data can be retrieved as this link selecting "Venus" as target and TIS as instrument channel. In addition, we have uploaded the complete calibrated MERTIS TIS data obtained during the second Venus flyby to figshare (https://doi.org/10.6084/m9.figshare.24476812.v1).

## Code availability

The computer codes used for radiative transfer calculations and retrieval procedures are direct implementations of published methods[2,21,22,25].

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

## Acknowledgements

We are grateful to the whole MERTIS engineering team including all industrial partners and to the ESA BepiColombo team that made these observations possible at all.

## Author contributions

J.H. led the study and wrote the manuscript. R.H. and G.A. performed the data analysis and modeling. M.d.A., A.M., J.H. and T.S. worked on the data calibration and data processing. A.M. performed observation planning. All authors participated in the interpretation of the results, in science data acquisition, mission operations, or project management, and/or contributed to discussion.

## Funding

## Competing interests

The authors declare no competing interests.
