## [Peer Review File · Nature Communications]

REVIEWER COMMENTS

Reviewer #1 (Remarks to the Author):

Comments on: "MERTIS observations of the atmosphere of Venus during the second BepiColombo flyby" by Jorn Helbert et al.

Thermal infrared spectra (TIS) in the spectral range 6.8-13.9 μm (720-1470 cm^{-1}) obtained by the MErcury Radiometer and Thermal Infrared Spectrometer (MERTIS) during the second Venus flyby of the ESA-JAXA BepiColombo mission were presented and analysed in this manuscript.

It is noteworthy that measurements with the MERTIS hyperspectral channel (MERTIS-TIS) are the first spectrally resolved observations of Venus in the thermal spectral range longward of 5 μm since the Venera-15 Fourier spectrometer experiment FS-1/4 in 1983.

The authors clearly shown that mesospheric parameters of Venus like temperature altitude profiles and cloud properties could be reliably determined from BepiColombo MERTIS spectra recorded during flyby 2.

The authors could present evidence of high consistency between the retrieved temperature profile at latitude 10N with the corresponding VIRA profile and the profile obtained from Venera-15 PMV data. Was also clear the high level of consistency between the retrieved cloud properties and cloud top altitudes.

I fully agree with the authors when they state that the results presented here contribute to our understanding of the atmosphere of Venus and its stability on timescales of decades. It is also evident that MERTIS is effective for the investigation of relatively cool objects based on hyperspectral observations. The present contribution is even more relevant regarding the advent of an increase of the space-based Venus' research in the near future (EnVision, VERITAS, DaVinci).

The description of the followed methodology in the data reduction, radiative transfer methods used and the process of obtaining the observations, was comprehensive, well explained and extremely careful. Which allowed me to fully understand the several steps that conducted to the results here presented. The related interpretation and conclusions are, in my opinion, solid and sturdy and are quite sustained by the results here presented.

In my opinion, the paper is strongly acceptable for publication after some very minor revision, mostly in the field of typo. Hereafter my remarks:

The second Venus flyby (FB2) on which we focus here took place on August 10, 2021

MERTIS data analyses that are presented here are based on Venus Flyby 2 measurements

Page 2, figure 1: typo in the legend of "Latitudes" please correct inserting the symbol for degrees.

Page 4, line 9: typo in the acronyms: might be "MERTIS-TIS"

Page 4, line 14-15: Regarding the statement: "Note that atmospheric emissions in the thermal spectral range longward of 7 μm are not affected by solar radiation.", consider to insert a reference that sustain this claim.

Page 7, figure 5B: Again in the figure legend there is a typo in the symbol: the infinite symbol is presented in place of the degree symbol, please correct it.

Page 11, line 13: Where is said "...and the cloud top altitude are 28.34 and 66.64 km." please consider to add to that sentence ", at the latitudes range sensed.". For high latitudes there is a "compression" of the cloud layer and the cloud tops is reached at sensible lower altitudes than at lower latitudes (see Ignatiev N., et al, 2009, as an example).

I congratulate the authors for this extremely relevant work. Surely the planetary sciences community would be quite grateful to this team, since it is obvious the challenging project of using MERTIS in the scope of this study and all the related huge effort required.

Pedro Machado

Reviewer #2 (Remarks to the Author):

General:

This paper presents results from the thermal infrared observation conducted by the MERTIS spectrometer onboard BepiColombo spacecraft during the second Venus flyby. The observation and analysis procedures are described in detail. The obtained spectra are compared with the spectra obtained by Venera-15 to show the validity of the new measurement. The vertical profile of the atmospheric temperature was retrieved and was shown to be consistent with the previous observations, although error analysis, which is essential for studies utilizing retrieval, is not described. This paper has demonstrated that a spectrometer using an uncooled microbolometer is useful for Venus observations. On the other hand, scientific achievements are not adequately described.

Specific comments:

- p.3, l.15: The meaning of "T_B" should be described.
- p.3, l.15: The definition of "observation angle" should be given.
- Figure 2A: Given the apparent large noise level of the original individual spectra, the magnitude of the error for the averaged spectrum should be mentioned in the figure or in the main text. The label for the vertical axis is missing.
- p.5, l.3: It should be stated that the cloud model is described in the Methods section.
- p.6, l.3: The meaning of "simulated PMV brightness temperature spectra" is unclear. I guess this is a forward model calculation using the retrieved atmospheric quantities.
- p.6 l.13: The definition of "cloud mode factor MF_{1/2}" is unclear. Does this mean M₁ and M₂ ? If so, it is unclear why only Mode 1 and Mode 2 are adjusted.
- p.6, l.27: The meaning of "Delta T_B" should be described.
- p.7, l.4: The meaning of "BT" should be described.
- p.7, l.4: The meaning of "the only condition in the retrieval procedure" is unclear.
- p.7: It is mentioned that the obtained result contributes to our understanding of the stability of the Venusian atmosphere, but this argument is too abstract. The retrieval error should be compared with expected long-term variation ranges to show that the retrieval is precise enough. For example, Tellmann et al. (2009, JGR 114, E00B36) compares the Venus Express radio occultation temperatures with the Pioneer Venus's observations, providing possible variation ranges.

Reviewer #3 (Remarks to the Author):

This paper describes the spectral observations of Venus taken between 6.8-13.9 μm , a range in which the radiance from Venus is much lower than the temperatures on Mercury for which the instrument was designed. The paper details the procedure used to show that the retrieved temperature profiles are consistent with the known thermal structure of Venus indicating that the MERTIS instrument is able to provide data even at low signal level. There is little new in the science results described in the paper, so the results are not particularly newsworthy nor does the single profile provided add significantly to the thermal structure observations.

The paper is generally well written although there are some places where the text can be improved. Detailed comments are given below.

Page/Line # Comment

2/5 “..will record the emissivity spectra” – the TIR will detect radiation, not emissivity, so this wording is not quite appropriate. The spectral variation of emissivity is inferred not recorded.

2/7 “TIR is implemented by an in-plane separation arrangement” – conveys nothing to me. Separation of what from what? A figure would be useful or more text describing the arrangement.

2/37 “The TIR was operating at full spatial resolution (100 pixels)” – 100 pixels conveys little useful information to the lay reader without knowing more about the optical design and sensor specifications of the instrument. Suggest deleting it.

Figure 1 Legend inside the figure states “94.005 spectra” – is 94,005 intended here? Even more confusing, line 38 states that more than 900,000 spectra were obtained.

Figure 2 What main message does this figure showing TIR and PMV spectra (taken decades earlier) intends to convey? The spectral ranges are different? The large “noise” in the TIR spectra? How many spectra are shown in A and B? Only a few colors can be recognized. Are the numbers the same for both TIR and PMV? It may be more useful to show the standard deviation and average values for both instead of including numerous spectra.

3/23 Perhaps showing the histogram of Uobs in 10° latitude as a sub-plot in Figure 1 would be useful.

5/2 “cloud magnitude” is not a commonly used term in this context- perhaps the authors mean “cloud optical thickness”?

Figure 3A It would be useful to include a comment about the differences in the half widths of the TIR and PMV spectrometers.

5/28 what is meant by “repeated retrievals from PMV measurements”? Please clarify.

6/7 "Initial profile of VIRA" - assume the VIRA profile was used to calculate the simulated radiances? If so, it could be attributed as such rather than "initial".

6/9-13 Suggest including the differences in the retrieved and the VIRA profile on secondary vertical axis in Figures 4A and 4B.

9/41-42 How was the pointing accuracy from the SPICE kernels verified? Typically, the pointing is improved by using images, but it is not obvious what the approach used here as it is not described.

Response to Reviewer #1 Comments

Please note: Page and line numbers refer to the first version of the paper. New line numbers are additionally given in blue color, therefore.

A brief introduction about the atmosphere of Venus is now included at the beginning of the Introduction section. The arrangement of paragraphs in this section was changed. Additional paragraphs come from the old Section "Main". Other parts from that section were shifted to the Results section.

The implications and the importance of your results are now summarized in the last five paragraphs in the Results section (Lines 288-324).

Reviewer #1 (Remarks to the Author):

Comments on: "MERTIS observations of the atmosphere of Venus during the second BepiColombo flyby" by Jorn Helbert et al.

Thermal infrared spectra (TIS) in the spectral range 6.8-13.9 μm (720-1470 cm^{-1}) obtained by the MErcury Radiometer and Thermal Infrared Spectrometer (MERTIS) during the second Venus flyby of the ESA-JAXA BepiColombo mission were presented and analysed in this manuscript.

It is noteworthy that measurements with the MERTIS hyperspectral channel (MERTIS-TIS) are the first spectrally resolved observations of Venus in the thermal spectral range longward of 5 μm since the Venera-15 Fourier spectrometer experiment FS-1/4 in 1983.

The authors clearly shown that mesospheric parameters of Venus like temperature altitude profiles and cloud properties could be reliably determined from BepiColombo MERTIS spectra recorded during flyby 2.

The authors could present evidence of high consistency between the retrieved temperature profile at latitude 10N with the corresponding VIRA profile and the profile obtained from Venera-15 PMV data. Was also clear the high level of consistency between the retrieved cloud properties and cloud top altitudes.

I fully agree with the authors when they state that the results presented here contribute to our understanding of the atmosphere of Venus and its stability on timescales of decades. It is also evident that MERTIS is effective for the investigation of relatively cool objects based on hyperspectral observations. The present contribution is even more relevant regarding the advent of an increase of the space-based Venus' research in the near future (EnVision, VERITAS, DaVinci).

The description of the followed methodology in the data reduction, radiative transfer methods used and the process of obtaining the observations, was comprehensive, well explained and extremely careful. Which allowed me to fully understand the several steps that conducted to the results here presented. The related interpretation and conclusions are, in my opinion, solid and sturdy and are quite sustained by the results here presented.

In my opinion, the paper is strongly acceptable for publication after some very minor revision, mostly in the field of typo. Hereafter my remarks:

The second Venus flyby (FB2) on which we focus here took place on August 10, 2021

This is mentioned in the introduction on P2L24 (Lines 57-58). We added also added it in the observations section in Lines 123-124 for clarity

MERTIS data analyses that are presented here are based on Venus Flyby 2 measurements

This paragraph was shifted to the Results/Observation section (Lines 122-121).

Page 2, figure 1: typo in the legend of "Latitudes" please correct inserting the symbol for degrees.

This was a mistake while embedding the figure in the document. The figure is correct and now is also displayed correctly in the embedded version.

Page 4, line 9: typo in the acronyms: might be "MERTIS-TIS"

This has been corrected in line 176

Page 4, line 14-15: Regarding the statement: "Note that atmospheric emissions in the thermal spectral range longward of 7 μm are not affected by solar radiation.", consider to insert a reference that sustain this claim.

A reference is now added to the text (line 182).

Page 7, figure 5B: Again in the figure legend there is a typo in the symbol: the infinite symbol is presented in place of the degree symbol, please correct it.

As for figure 1 this seems to be an issue with embedding the figures in the review version. The actual figure has the correct version and the embedded version is correct now too.

Page 11, line 13: Where is said "...and the cloud top altitude are 28.34 and

66.64 km." please consider to add to that sentence ", at the latitudes range sensed.". For high latitudes there is a "compression" of the cloud layer and the cloud tops is reached at sensible lower altitudes than at lower latitudes (see Ignatiev N., et al, 2009, as an example).

We fully agree with your assessment that the cloud top altitude can depend on latitude. This was one of the main results presented by Haus et al. (2013, Fig. 32). In the given context, however, this is not applicable. Decreasing top altitudes at higher latitudes are caused by decreasing mode factors $MF_{1/2}$. Here we are using the cloud standard model where the mode factors MF_j are unity. Therefore, the given values are correct and do not depend on latitude. Line 426

I congratulate the authors for this extremely relevant work. Surely the planetary sciences community would be quite grateful to this team, since it is obvious the challenging project of using MERTIS in the scope of this study and all the related huge effort required.

Response to Reviewer #2 Comments

Please note: Page and line numbers refer to the first version of the paper. New line numbers are additionally given in blue color, therefore.

A brief introduction about the atmosphere of Venus is now included at the beginning of the Introduction section. The arrangement of paragraphs in this section was changed. Additional paragraphs come from the old Section “Main”. Other parts from that section were shifted to the Results section.

The implications and the importance of your results are now summarized in the last five paragraphs in the Results section (Lines 287-322).

Reviewer #2 (Remarks to the Author):

General:

This paper presents results from the thermal infrared observation conducted by the MERTIS spectrometer onboard BepiColombo spacecraft during the second Venus flyby. The observation and analysis procedures are described in detail. The obtained spectra are compared with the spectra obtained by Venera-15 to show the validity of the new measurement. The vertical profile of the atmospheric temperature was retrieved and was shown to be consistent with the previous observations, although error analysis, which is essential for studies utilizing retrieval, is not described. This paper has demonstrated that a spectrometer using an uncooled microbolometer is useful for Venus observations. On the other hand, scientific achievements are not adequately described.

error analysis, which is essential for studies utilizing retrieval, is not described

Text added near Fig.2, Lines 167-171: Since the instrument under Venus conditions has worked at its limits, single spectra are very noisy and inappropriate for use in retrieval procedures. The average spectrum used for the retrieval is based on almost 13000 noise single spectra as can be seen in Figure 2a. Therefore, detailed error analyses of retrieval results are not possible. Averaging leads to a major reduction of measurement errors. This is proven by the structure of the averaged spectrum, which is well comparable with PMV.

We added Figure 6 which shows a comparison between VIRA, PMV and VIRTIS (using the 4.3 μm band) profiles with the MERTIS derived profile

scientific achievements are not adequately described

The implications and the importance of your results are now summarized in the last five paragraphs in the Results section (Lines 280-316).

Specific comments:

- p.3, l.15: *The meaning of "T_B" should be described.*

We added brightness temperature (T_B) in line 154

- p.3, l.15: *The definition of "observation angle" should be given.*

we added in Line 147 observation angle (often denoted as emission angle)

- *Figure 2A: Given the apparent large noise level of the original individual spectra, the magnitude of the error for the averaged spectrum should be mentioned in the figure or in the main text.*

We added the standard deviation to Figure 2a to show the noise level in the individual spectra.

The label for the vertical axis is missing.

To optimize the use of space and because it is the same for Fig.2A and 2B we have given the label for the vertical axes in the middle of the two figures (cf. Fig. 3).

- p.5, l.3: *It should be stated that the cloud model is described in the Methods section.*

We added a sentence in Line 197: The used cloud model is described in the Methods section.

- p.6, l.3: *The meaning of "simulated PMV brightness temperature spectra" is unclear. I guess this is a forward model calculation using the retrieved atmospheric quantities.*

To clarify this we added a sentence in Lines 229-230: "Simulated spectra are obtained from forward modeling using the retrieved parameters as input quantities."

- p.6 l.13: *The definition of "cloud mode factor MF_{1/2}" is unclear. Does this*

mean M_1 and M_2 ? If so, it is unclear why only Mode 1 and Mode 2 are adjusted.

To clarify this we added in Lines 239-242 some explanations:

“Retrieved cloud mode factors of modes 1 and 2 ($MF_{1/2}$, see Methods section) and.....”

“Factors of modes 3 and 4 cannot be retrieved in the used spectral ranges.”

- p.6, l.27: The meaning of "Delta T_B " should be described.

We added in Line 257 “Deviations of brightness temperatures ($|\Delta T_B|$) are mostly.....”

- p.7, l.4: The meaning of "BT" should be described.

This was a mix up of variable naming – we corrected this in Line 267 - BT substituted by T_B

- p.7, l.4: The meaning of "the only condition in the retrieval procedure" is unclear.

To clarify this we added in Line 268 one word: “..is the only **decisive** condition..”

- p.7: It is mentioned that the obtained result contributes to our understanding of the stability of the Venusian atmosphere, but this argument is too abstract. The retrieval error should be compared with expected long-term variation ranges to show that the retrieval is precise enough. For example, Tellmann et al. (2009, JGR 114, E00B36) compares the Venus Express radio occultation temperatures with the Pioneer Venus's observations, providing possible variation ranges.

- p.7: It is mentioned that the obtained result contributes to our understanding of the stability of the Venusian atmosphere, but this argument is too abstract. The retrieval error should be compared with expected long-term variation ranges to show that the retrieval is precise enough. For example, Tellmann et al. (2009, JGR 114, E00B36) compares the Venus Express radio occultation temperatures with the Pioneer Venus's observations, providing possible variation ranges.

We added Figure 6 which shows a comparison between VIRA, PMV and VIRTIS (using the 4.3 μm band) profiles with the MERTIS derived profile

Response to Reviewer #3 Comments

Please note: Page and line numbers refer to the first version of the paper. New line numbers are additionally given in blue color, therefore.

A brief introduction about the atmosphere of Venus is now included at the beginning of the Introduction section. The arrangement of paragraphs in this section was changed. Additional paragraphs come from the old Section "Main". Other parts from that section were shifted to the Results section.

The implications and the importance of your results are now summarized in the last five paragraphs in the Results section (Lines 288-324).

This paper describes the spectral observations of Venus taken between 6.8-13.9 μm , a range in which the radiance from Venus is much lower than the temperatures on Mercury for which the instrument was designed. The paper details the procedure used to show that the retrieved temperature profiles are consistent with the known thermal structure of Venus indicating that the MERTIS instrument is able to provide data even at low signal level. There is little new in the science results described in the paper, so the results are not particularly newsworthy nor does the single profile provided add significantly to the thermal structure observations.

We try to address this now in several places;
see Results/Observations section, Lines 122-123.

It was written there: Due to the spacecraft trajectory and the comparatively low distance to Venus (6300 km on average) most data were acquired at latitudes between 8 and 14°N and at local times between 7 and 15 h.

The following sentence was added: Lines 124-126

Therefore, FB2 data contain limited information on Venus thermal structure on global scale from the outset, but nevertheless their analysis is important to validate the above discussed approach (direct comparison with PMV, preparation of FB1 data analyses)

See also Lines 284-321. The implications and the importance of your results are now summarized in the last five paragraphs in the Results section

The paper is generally well written although there are some places where the text can be improved. Detailed comments are given below.

Page/Line # Comment

2/5 “..will record the emissivity spectra” – the TIR will detect radiation, not emissivity, so this wording is not quite appropriate. The spectral variation of emissivity is inferred not recorded.

We correct this in Lines 55-56, Modification is:will record dayside radiance spectra of Mercury to infer surface emissivity characteristics, whereas.

2/7 “TIR is implemented by an in-plane separation arrangement” – conveys nothing to me. Separation of what from what? A figure would be useful or more text describing the arrangement.

We added a sentence to explain this better (Lines 68-69) as well as a reference with additional details and figures: “In this configuration the two radiometer detector lines form the slit of the TIS channel which is an imaging spectrometer (Hiesinger et al. 2020)”

2/37 “The TIR was operating at full spatial resolution (100 pixels)” – 100 pixels conveys little useful information to the lay reader without knowing more about the optical design and sensor specifications of the instrument. Suggest deleting it.

We clarified this statement by adding a sentence (Lines 134-135): “The TIS channel was working at full spatial resolution without any spatial binning providing 100 pixels across the track, and ...”

Figure 1 Legend inside the figure states “94.005 spectra” – is 94,005 intended here? Even more confusing, line 38 states that more than 900,000 spectra were obtained.

We corrected the German to English notation 94.000 → 94,000

Regarding the statement that more than 900,000 were obtained: This is the correct number of all recorded spectra. After removal of very weak spectra (see data selection description) the remaining number is about 94,000.

Figure 2 What main message does this figure showing TIR and PMV spectra (taken decades earlier) intends to convey? The spectral ranges are different? The large “noise” in the TIR spectra? How many spectra are shown in A and B? Only a few colors can be recognized. Are the numbers the same for both TIR and PMV? It may be more useful to show the standard deviation and average values for both instead of including numerous spectra.

The main message of this figure is that those are currently the only hyperspectral TID data sets of Venus taken by a spacecraft in close proximity. The figure shows the differences between data taken with an instrument designed for Mercury science using a simple grating and a Fourier Spectrometer flown “decades earlier”. This does include the message that “The spectral ranges are different?” and that despite a limited wavelength coverage in MERTIS compared to PMV still useful temperature profiles can be derived.

We show for MERTIS 200 out of all 94000 spectra purely for visualisation, For B all 15 existing spectra are shown. Therefore, the number of spectra in A and B are also not the same.

The colors of the individual spectra are only meant as a guide to the eye helping to see how noisy a single spectrum is.

We agree that the standard deviation might be helpful as an additional information and therefore we added it in Figure 2 a as cyan dashed lines. We still kept the actually spectra to show the noisy nature of the individual spectra.

3/23 Perhaps showing the histogram of Uobs in 10° latitude as a sub-plot in Figure 1 would be useful.

A very good suggestion – Histograms Number of spectra as function of observation angle are now inserted

5/2 “cloud magnitude” is not a commonly used term in this context- perhaps the authors mean “cloud optical thickness”?

Done. Line 196-197 Cloud magnitude → cloud optical thickness

Figure 3A It would be useful to include a comment about the differences in the half widths of the TIR and PMV spectrometers.

We added in Lines 194-195 the following text: The spectral resolution of PMV data is 6.3 cm^{-1} . The MERTIS resolution of 90 nm corresponds to 18 cm^{-1} near $7 \mu\text{m}$ and 5 cm^{-1} near $14 \mu\text{m}$.

5/28 what is meant by “repeated retrievals from PMV measurements”? Please clarify.

Done. Line 222 “Repeated” deleted

6/7 "Initial profile of VIRA" - assume the VIRA profile was used to calculate the simulated radiances? If so, it could be attributed as such rather than "initial".

The VIRA profile serves as the initial profile in the retrieval algorithm only. Simulated (retrieved) brightness temperatures are calculated using the retrieved temperature profile that may or may not, more or less, deviate from the initial one. Therefore we feel the use of "initial" is correct here

6/9-13 Suggest including the differences in the retrieved and the VIRA profile on secondary vertical axis in Figures 4A and 4B.

Very good suggestion - added in Fig. 4B as well as Fig. 5B.

9/41-42 How was the pointing accuracy from the SPICE kernels verified?

Typically, the pointing is improved by using images, but it is not obvious what the approach used here as it is not described.

Additional details provided in (Lines 365-367) "This includes validation of the internal alignment using the onboard 700K blackbody emitter as well as validation of the external alignment based on imaging of the Moon during the Moon flyby."

REVIEWER COMMENTS

Reviewer #1 (Remarks to the Author):

The revised version of the manuscript addresses all my previous comments in a satisfactory way.

In my opinion this paper is acceptable for publication in the present form.

Pedro Machado

Reviewer #2 (Remarks to the Author):

Statements about the importance of this observation have been added to the manuscript. However, my understanding is that the main scientific finding of this study is the stability of atmospheric temperatures over decades. If so, more discussion of long-term variability, citing previous studies, would be needed. For example, Tellmann et al. (2009) compared the Venus Express radio occultation results with PVO, VIRA and VIRA-2 temperatures, and Ando et al. (2020, Sci Rep 10:3448) compared Akatsuki radio occultation results with Venus Express, Magellan, VIRA and VIRA-2. Evidence of long-term variation has been obtained in albedo and mean winds as shown by Lee et al. (2019, Astron J 158:126) and Peralta et al. (2018, ApJ Supp 239:29) and others.

Error analysis of retrieved temperatures is still lacking. It is stated that "detailed error analyses of retrieval results are not possible". This is understandable to some extent, but at least the absolute radiometric calibration error of the instrument may cause a systematic error in the retrieved temperature. It is then recommended that a sensitivity analysis using simulated spectra be performed. Without such an analysis, it is not possible to determine whether the difference of a few K between the present and previous observations is within the error range, although the reviewer agrees with the conclusion that the temperature change is probably within this range.

Reviewer #3 (Remarks to the Author):

Please add the temperature differences between the different profiles shown in Figure 6 as you have done for figures 4b and 5b. As presented it is not easy to discern how small or large the differences are between VIRTIS, VIRA, PMV and MERTIS profiles.

Response to REVIEWER COMMENTS

Reviewer #1 (Remarks to the Author):

The revised version of the manuscript addresses all my previous comments in a satisfactory way. In my opinion this paper is acceptable for publication in the present form.

Pedro Machado

Reviewer #2 (Remarks to the Author):

Statements about the importance of this observation have been added to the manuscript. However, my understanding is that the main scientific finding of this study is the stability of atmospheric temperatures over decades. If so, more discussion of long-term variability, citing previous studies, would be needed. For example, Tellmann et al. (2009) compared the Venus Express radio occultation results with PVO, VIRA and VIRA-2 temperatures, and Ando et al. (2020, Sci Rep 10:3448) compared Akatsuki radio occultation results with Venus Express, Magellan, VIRA and VIRA-2. Evidence of long-term variation has been obtained in albedo and mean winds as shown by Lee et al. (2019, Astron J 158:126) and Peralta et al. (2018, ApJ Supp 239:29) and others.

We are grateful for pointing out these very relevant references. We have updated the manuscript to discuss the complementarity and uniqueness of the observations presented here in the context of the previous measurements especially using radio occultation methods.

Error analysis of retrieved temperatures is still lacking. It is stated that "detailed error analyses of retrieval results are not possible". This is understandable to some extent, but at least the absolute radiometric calibration error of the instrument may cause a systematic error in the retrieved temperature. It is then recommended that a sensitivity analysis using simulated spectra be performed. Without such an analysis, it is not possible to determine whether the difference of a few K between the present and previous observations is within the error range, although the reviewer agrees with the conclusion that the temperature change is probably within this range.

We agree that the absolute radiometric calibration of the MERTIS instrument was insufficiently discussed. We have extended the method section for this purpose adding a discussion of the inflight calibration verification that is routinely performed as part of any MERTIS observation sequence. This shows that the error in the absolute calibration is significantly smaller than the deviation observed. A detailed sensitivity analysis using simulated spectra is beyond the scope of this paper and would be part of a follow-up publication on the data from the first Venus flyby which covers a much larger range of viewing geometries and latitudes.

Reviewer #3 (Remarks to the Author):

Please add the temperature differences between the different profiles shown in Figure 6 as you have done for figures 4b and 5b. As presented it is not easy to discern how small or large the differences are between VIRTIS, VIRA, PMV and MERTIS profiles.

We have updated Figure 5 to include the VIRTIS profile and the differences as was done in figures 4b and 5b. Figure 6 was removed accordingly.

REVIEWER COMMENTS

Reviewer #2 (Remarks to the Author):

I appreciate that the authors included descriptions about in-flight calibration in the methods section. If I am correct, this calibration ensures that the brightness temperature spectrum has an absolute error of 0.03 K or less. It gives an impression that the effect of absolute calibration errors on the atmospheric temperatures retrieved is smaller than the deviations discussed. If this is the case, the authors should provide a quantitative description of the absolute error of the brightness temperature spectrum in the main text, rather than just in the methods section. They should also argue that the absolute error in the retrieved atmospheric temperatures will have a similar magnitude. Including such details is necessary, even if a full sensitivity analysis is not conducted.

Response to REVIEWER COMMENTS

Reviewer #2 (Remarks to the Author):

I appreciate that the authors included descriptions about in-flight calibration in the methods section. If I am correct, this calibration ensures that the brightness temperature spectrum has an absolute error of 0.03 K or less. It gives an impression that the effect of absolute calibration errors on the atmospheric temperatures retrieved is smaller than the deviations discussed. If this is the case, the authors should provide a quantitative description of the absolute error of the brightness temperature spectrum in the main text, rather than just in the methods section. They should also argue that the absolute error in the retrieved atmospheric temperatures will have a similar magnitude. Including such details is necessary, even if a full sensitivity analysis is not conducted.

We added a more detailed explanation in this revision concerning several uncertainties in our measurements that we cannot quantify as the measurements during the flybys have been performed by ESA on a best effort base. As described in the method section MERTIS theoretically has an accuracy in the range of 0.03K based on its internal calibration. We will be able to achieve an overall accuracy in that range at Mercury when using the MERTIS planetary baffle, which is optimized to reject all out-of-field radiation from a source larger than the field of view of the instrument. For the Venus observations, however, the instrument had to observe using the space port. This port has a simpler, asymmetrically shaped baffle, which means that stray light from the extended disk of Venus must be considered as an additional source of uncertainty. Since no in-flight calibration of the space port could be performed for the Venus flyby, we chose the data clustering described in the paper to minimize this error for temperature retrieval. As instrument team we have been following a very careful approach for addressing uncertainties in this very specific kind of measurements without going into an over interpretation of the data. We have been clarifying in the revision that our results do not exclude temperature variability in the temperature profiles of Venus that could be observable with radio occultation methods.

REVIEWERS' COMMENTS

Reviewer #2 (Remarks to the Author):

The reviewer's primary concern that the absolute error in the retrieved temperature profile is unclear was solved to some extent. It is understandable that the observation condition during the Venus flyby did not allow precise calibration, and thus, the absolute error might differ from that in an ideal situation. Now, the readers can understand both the possible accuracy in a perfect case and the limitations of the result. Considering the importance of the measurement technique and the significance of the comparison between different missions, the paper is considered acceptable for publication.

Minor comments:

Line 105: The word "PMV" first appears here before its definition is given on line 112.

Line 112: What does "This instrument" mean should be made more explicit. I thought it meant MERTIS-TIS when I read it first.

Response to REVIEWER COMMENTS

Reviewer #2 (Remarks to the Author):

The reviewer's primary concern that the absolute error in the retrieved temperature profile is unclear was solved to some extent. It is understandable that the observation condition during the Venus flyby did not allow precise calibration, and thus, the absolute error might differ from that in an ideal situation. Now, the readers can understand both the possible accuracy in a perfect case and the limitations of the result. Considering the importance of the measurement technique and the significance of the comparison between different missions, the paper is considered acceptable for publication.

Minor comments:

Line 105: The word "PMV" first appears here before its definition is given on line 112. We added the definition of the word PMV at its first occurrence.

Line 112: What does "This instrument" mean should be made more explicit. I thought it meant MERTIS-TIS when I read it first. We replaced "This instrument" with PMV and added PMV in the sentence before.